# Research

ecology, environmental science

mesopelagic fish, Southern Ocean, biomass, acoustics, target strength

**Author for correspondence:**
Tracey Dornan
e-mail: tarna70@bas.ac.uk

# Large mesopelagic fish biomass in the Southern Ocean resolved by acoustic properties

Tracey Dornan[1,2], Sophie Fielding[1], Ryan A. Saunders[1] and Martin J. Genner[2]

[1]British Antarctic Survey, High Cross, Madingley Road, Cambridge CB3 0ET, UK
[2]School of Biological Sciences, University of Bristol, Life Sciences Building, 24 Tyndall Avenue, Bristol BS8 1TQ, UK

(iD) TD, 0000-0001-8265-286X

The oceanic mesopelagic zone, 200–1000 m below sea level, holds abundant small fishes that play central roles in ecosystem function. Global mesopelagic fish biomass estimates are increasingly derived using active acoustics, where echosounder-generated signals are emitted, reflected by pelagic organisms and detected by transducers on vessels. Previous studies have interpreted a ubiquitous decline in acoustic reflectance towards the Antarctic continent as a reduction in mesopelagic fish biomass. Here, we use empirical data to estimate species-specific acoustic target strength for the dominant mesopelagic fish of the Scotia Sea in the Southern Ocean. We use these data, alongside estimates of fish relative abundance from net surveys, to interpret signals received in acoustic surveys and calculate mesopelagic biomass of the broader Southern Ocean. We estimate the Southern Ocean mesopelagic fish biomass to be approximately 274 million tonnes if Antarctic krill contribute to the acoustic signal, or 570 million tonnes if mesopelagic fish alone are responsible. These quantities are approximately 1.8 and 3.8 times greater than previous net-based biomass estimates. We also show a peak in fish biomass towards the seasonal ice-edge, corresponding to the preferred feeding grounds of penguins and seals, which may be at risk under future climate change scenarios. Our study provides new insights into the abundance and distributions of ecologically significant mesopelagic fish stocks across the Southern Ocean ecosystem.

## 1. Introduction

Mesopelagic fish inhabit the twilight zone of the world's oceans, 200–1000 m below sea level. These typically small (less than 20 cm) fish are globally the most abundant vertebrates, and their communities are often dominated by lanternfish (family Myctophidae) in terms of number and biomass [1]. Mesopelagic fish play a vital role in biogeochemical cycling through extensive diel vertical migration, where large proportions of the community ascend to feed in nutrient-rich surface waters under the cover of darkness and actively sequester carbon when respiring at depth [2,3]. In the Southern Ocean, myctophids are fundamental to Antarctic food webs. They are key prey items of many higher predators, including king penguin [4], Antarctic fur seal [5] and elephant seal [6,7]. Myctophids are also consumers of a wide variety of zooplankton including the keystone Antarctic krill *Euphausia superba* [8]. There has been increased interest in the opportunities the mesopelagic zone has to offer, as we consider new ways to sustainably meet the needs of the growing human population [3,9]. However, large-scale biomass estimates for the Southern Ocean are rare, which limits our ability to quantify the importance of mesopelagic fish in ecosystem functioning and carbon sequestration.

Global net-based biomass estimates for mesopelagic fish are in the region of approximately 1000 million tonnes (Mt) [1], with an estimate of 70–191 Mt in the Southern Ocean [10]. While net sampling provides valuable data on the life history and relative abundance of mesopelagic fishes within their communities, nets can only ever sample a tiny proportion of the ocean's interior [11]. Given patchy species distributions and net avoidance behaviour, net-based assessments are likely to underestimate overall fish biomass [11,12].

Active acoustics have the ability to sample the water column at unrivalled spatial resolution, with studies suggesting that global mesopelagic fish biomass may be an order of magnitude higher than net-based estimates [13]. Acoustic methods also show declines in mesopelagic backscatter towards the poles, which has been interpreted as lower fish densities in the Southern Ocean relative to temperate latitudes [14,15]. However, those declines in backscatter are associated with a change in mesopelagic fish community composition, with the dominant species in the warmer lower latitudes having highly reflective gas-filled swimbladders, and species at the colder higher latitudes lacking gas-filled swimbladders, which contributes to the reduction of acoustic signal towards Antarctica [16].

To quantitatively interpret acoustic backscatter from mesopelagic fish communities, we require data on biological community composition and the acoustic reflective properties of species present [17,18]. Information on swimbladder gas and tissue density are required to model the acoustic reflectance or 'target strength' (TS) of species. However, the absence of such data for Antarctic mesopelagic fish has hampered our ability to develop species-specific TS models. Moreover, Southern Ocean mesopelagic fish species have a high degree of ontogenetic swimbladder variability, with some losing the gas component in adulthood [16,19]. Since gas in a swimbladder can account for up to 95% of the backscatter from a fish [20], loss of gas with increasing body size results in a nonlinear effect in a species' backscattering response, where larger fish may return a considerably lower acoustic signal than smaller gas-bearing individuals. Partitioning observed acoustic backscatter among species and life stages is, therefore, required to interpret acoustic data and derive biomass estimates.

Here, we develop the first taxon-specific TS estimates for key Southern Ocean mesopelagic fish species, using empirical and literature-derived information on tissue density and swimbladder gas content. We then use 6 years of acoustic transect data from the Scotia Sea and generalized additive mixed models (GAMMs) with environmental predictors, to predict large-scale patterns of acoustic backscatter for the Southern Ocean, which for the purposes of this study we define as the region south of 50° S. Finally, we assign proportions of this acoustic backscatter to individual species using mesopelagic fish assemblage data and unique TS values, enabling estimations of mesopelagic fish biomass for both the Scotia Sea and the wider Southern Ocean.

## 2. Methods

### (a) Acoustic surveys and net sampling in the Scotia Sea
As described in [16], we used a 38 kHz hull-mounted EK60 echosounder to collect acoustic data from the Scotia Sea during six research cruises (13 transects), spanning austral spring to autumn of 2006–2017. Acoustic transects crossed the major circumpolar fronts and water masses of the Southern Ocean (figure 1a, electronic supplementary material, table S1). Data were collected to 1000 m on all transects with the exception of JR161 ($n = 2$) and JR200 ($n = 3$), where data were collected to 800 m and 990 m, respectively. Data were calibrated, cleaned and processed in Echoview (v. 8.0.95, Echoview Software Pty Ltd, Hobart, Australia). Nautical area scattering coefficient (NASC, $m^2$ $nmi^{-2}$), a measure of mean water column backscatter, was depth integrated (17–1000 m or maximum available) in 1 km distance sampling units. Data collected over the shelf (bathymetry ≤ 1000 m) were excluded to restrict analysis to mesopelagic waters. See electronic supplementary material for a description of acoustic data cleaning and processing. All subsequent analyses were completed in R (v. 3.5.3) [21].

We also collected mesopelagic fish samples using a remotely operated opening and closing rectangular mid-water trawl system (RMT25), with apertures of 25 $m^2$ (mesh size: 8–4.5 mm) on five of the cruises, covering the same vertical resolution as the acoustic data (1000 m—surface, figure 1a). Samples were used to determine community composition and measure the morphological data required for TS modelling, including length–frequency, length–width ratio, length–weight regressions, tissue density and swimbladder condition. An overview of the morphological parameters assessed for TS modelling can be found in electronic supplementary material, table S2.

### (b) Target strength modelling
#### (i) Fish target strength models
TS (dB re 1 $m^2$) is the expected return acoustic signal from an organism given its unique scattering properties and the insonifying frequency. We modelled the TS at 38 kHz for 11 of the most abundant mesopelagic fish taxa, which accounted for greater than 94% of mesopelagic fish by abundance in RMT25 net samples [16]. For fish lacking a functional gas-filled swimbladder (*Gymnoscopelus braueri*, *Gymnoscopelus fraseri*, *Gymnoscopelus nicholsi*, large individuals of *Electrona antarctica* (SL ≥ 51.378 mm), *Bathylagus* spp., *Notolepis* spp. and *Cyclothone* spp.) [16], we used a simple fixed finite cylinder model [22], to represent the fish body, parameterized with locally derived measurements of fish tissue density, standard length and length to width ratio. The model, originally developed for zooplankton, considers cylinder tapering and is effective on a range of angles of orientation [22]. For fish with a gas-filled swimbladder (*Electrona carlsbergi*, small *E. antarctica* (less than 51.378 mm), *Krefftichthys anderssoni*, *Protomyctophum bolini* and *Protomyctophum tenisoni*) [16], we used a simple prolate spheroid model based on the volume of gas required to make a fish neutrally buoyant in surrounding seawater [23–25], which we modelled at 500 m depth. Full model descriptions can be found in the electronic supplementary material.

#### (ii) Tissue density measurements
Fish tissue densities were measured following the modified density bottle method in [17], by measuring the specific gravity (ratio of density relative to freshwater) of 81 fish from seven species at approximately 4°C, based on their availability in net samples (*E. antarctica* $n = 6$, *E. carlsbergi* $n = 7$, *G. braueri* $n = 19$, *G. fraseri* $n = 3$, *K. anderssoni* $n = 18$, *P. bolini* $n = 16$ and *Bathylagus* spp. $n = 12$), see electronic supplementary material, table S3. Fish known to have a gas-bearing swimbladder were dissected under water to exclude gas bubbles, and placed sequentially from low to higher density glycerol–seawater solutions to determine the point of neutral buoyancy and hence fish-specific gravity. Hydrometers were calibrated at 4°C post-cruise to convert

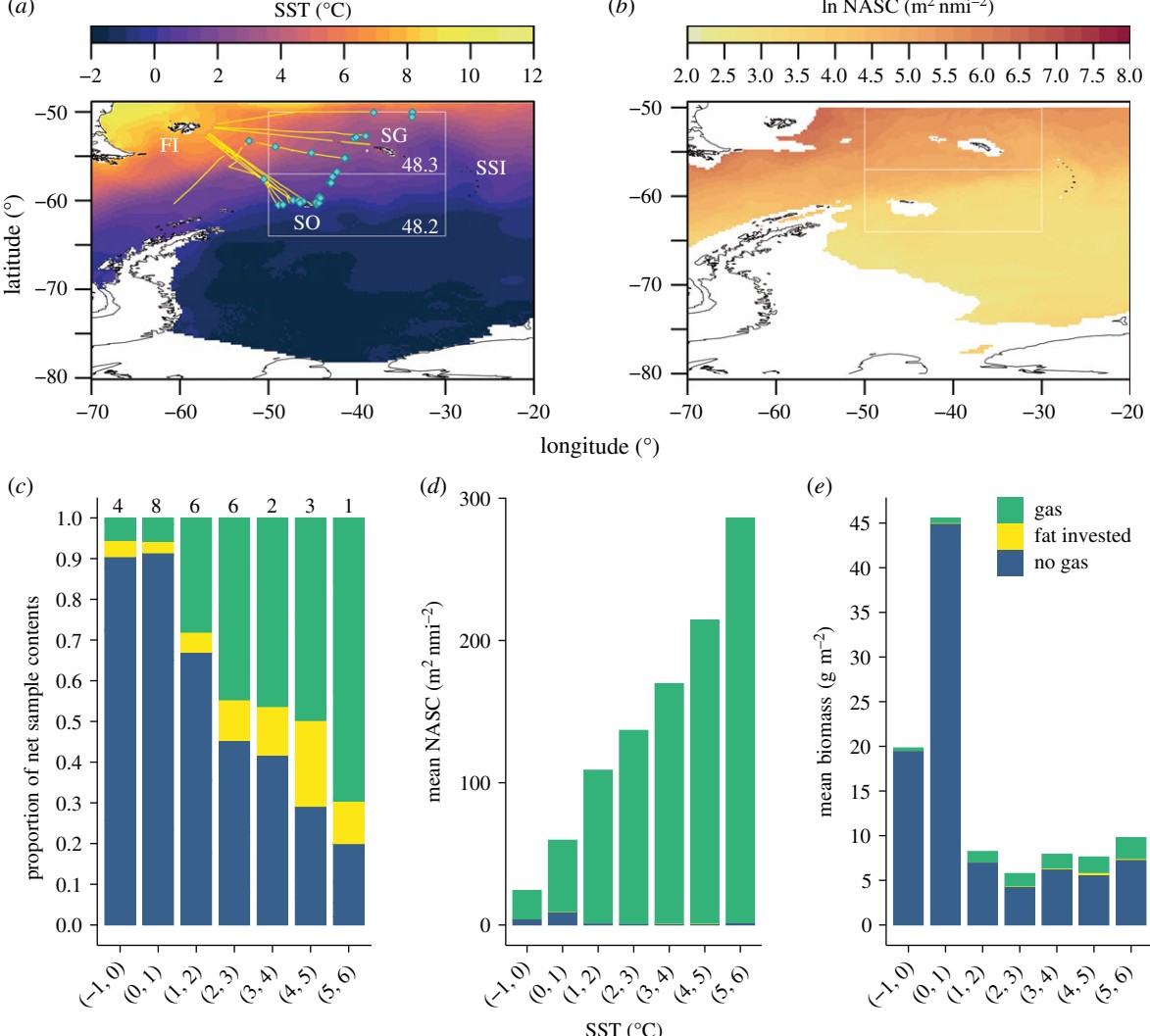

**Figure 1.** (*a*) Scotia Sea study area, showing sea surface temperature (SST) climatology. Yellow lines are 38 kHz acoustic transects, cyan diamonds indicate net sampling locations used to assess mesopelagic fish community composition and collect morphometric samples. White boxes are CCAMLR subareas 48.2/48.3, for which Scotia Sea fish biomass was calculated. FI, Falkland Islands; SG, South Georgia; SO, South Orkney Islands; SSI, South Sandwich Islands. (*b*) Predicted log$_e$ NASC for the Scotia Sea used to estimate fish biomass. (*c–e*) Bar plots illustrating the nonlinear effect of the presence or absence of gas on acoustic estimates of fish biomass. (*c*) Proportions of fish by swimbladder contents in net samples within 1°C SST groups. Numbers above bars indicate the number of total water column net samples in each group. (*d*) Acoustic contribution of fish species to mean predicted NASC in SST groups by swimbladder contents, using TS values for median length fish. (*e*) Relative proportions of gas and non-gas swimbladder fish species contributing to Southern Ocean mean biomass (g m$^{-2}$). Note that the relative proportion of gas-bearing species is highest at warmer temperatures (*c*) but overall contribute less to biomass estimate (*e*) because of the nonlinear effect of gas reflection on acoustic backscatter. (Online version in colour.)

specific gravity to density. For taxa where no samples were available for density measurement, each taxon was assigned the mean tissue density of all measured non-gas or gas-bearing taxa as appropriate (table 1). We used the mean tissue density of each taxon in all TS model calculations.

### (iii) Fish swimbladder gas volume

Fish wet weight and density values were used to calculate the theoretical equivalent gas volume that would be required for a fish to achieve neutral buoyancy in the density of surrounding seawater at atmospheric pressure (electronic supplementary material, table S3) [17]. Gas volume was calculated at atmospheric pressure, as fish densities were measured at atmospheric pressure, and it is assumed that the same gas volume will be maintained at depth [29]. The density of seawater was estimated as the mean surface to 1000 m density from the nearest conductivity temperature depth (CTD; Sea-Bird SBE911Plus) profile to the capture location of the fish (electronic supplementary material, table S4).

### (iv) Standard length and length to width ratio

Standard length (SL) measurements for all 11 study taxa (table 1) were taken from RMT25 day and night net catches (for net summaries see electronic supplementary material, table S5). Length–width ratio (LWR, table 1) was derived using laboratory measurements from cruises JR161 and JR177, for all species with the exception of *Bathylagus* spp., which was measured from digital photographs taken immediately after capture (*n* = 6), and *Notolepis* spp. and *Cyclothone* spp., which were assigned the mean LWR of all nine other taxa in the study.

### (v) Length weight regressions

With the exception of *Notolepis* spp., frozen fish from RMT25 day and night net catches were weighed, and used to generate taxon-specific length–weight regressions (electronic supplementary material, table S6). For *Notolepis* species, length–weight regression parameters were taken from FishBase (*Notolepis coatsi*) [30], with SL converted to total length, using the conversion factor for the closely related *Arctozenus risso* [30].

**Table 1.** Species morphological parameters and estimated target strength (TS) used in biomass estimation. KRA, *K. anderssoni*; PRM, *P. bolini*; PRE, *P. tenisoni*; ELC, *E. carlsbergi*; ELN_S, *E. antarctica* (less than 51.378 mm); ELN_L, *E. antarctica* (greater than 51.378 mm); GYR, *G. braueri*; GYF, *G. fraseri*; GYN, *G. nicholsi*; BAX, *Bathylagus* spp.; NOE, *Notolepis* spp.; YTX, *Cyclothone* spp.; KRI, *Euphausia superba* (krill). SL, standard length: N, number of individual fish contributing to SL measurements; P25% and P75% denote 25th and 75th percentiles, respectively. Mod is TS model where: PS, prolate spheroid model [23–25]; FC, finite cylinder model [22]; SDWBA, stochastic distorted wave bome approximation [26–28].

| taxon | $\rho_f$ kg m$^{-3}$ | LWR (n) | SL (mm) | | | | | mod | TS at SL (dB re 1 m$^2$) | | | |
|---|---|---|---|---|---|---|---|---|---|---|---|---|
| | | | N | P$_{25\%}$ | mean | median | P$_{75\%}$ | | P$_{25\%}$ | mean | median | P$_{75\%}$ |
| KRA | 1038.39 | 12.42 (365) | 964 | 35.00 | 45.09 | 43.00 | 58.50 | PS | −59.98 | −57.68 | −58.11 | −55.32 |
| PRM | 1061.67 | 11.36 (289) | 619 | 37.00 | 43.94 | 45.00 | 51.00 | PS | −55.60 | −54.09 | −53.89 | −52.79 |
| PRE | 1053.83[a] | 11.39 (58) | 145 | 33.00 | 39.33 | 42.00 | 47.00 | PS | −57.25 | −55.85 | −55.33 | −54.44 |
| ELC | 1061.44 | 9.13 (112) | 251 | 73.00 | 75.64 | 76.00 | 78.00 | PS | −49.39 | −49.10 | −49.06 | −48.85 |
| ELN_S | 1038.00 | 10.14 (1133) | 335 | 42.00 | 44.75 | 46.00 | 49.00 | PS | −58.28 | −57.70 | −57.45 | −56.87 |
| ELN_L | | | 2220 | 64.00 | 74.48 | 74.00 | 84.00 | FC | −80.99 | −78.13 | −78.25 | −76.00 |
| GYR | 1028.94 | 11.91 (484) | 1493 | 67.00 | 84.22 | 86.00 | 103.00 | FC | −84.69 | −80.27 | −79.89 | −76.71 |
| GYF | 1064.14 | 10.86 (66) | 118 | 59.00 | 65.46 | 66.00 | 76.00 | FC | −79.44 | −77.37 | −77.21 | −74.51 |
| GYN | 1043.38[b] | 9.51 (46) | 100 | 116.00 | 122.68 | 137.00 | 149.00 | FC | −69.65 | −69.11 | −68.33 | −68.10 |
| BAX | 1037.05 | [c]8.08 (6) | 1578 | 76.00 | 96.19 | 94.00 | 114.00 | FC | −75.09 | −71.88 | −72.15 | −70.41 |
| NOE | 1043.38[b] | [d]10.53 (—) | 185 | 63.75 | 76.07 | 72.00 | 83.00 | FC | −80.57 | −77.21 | −78.23 | −75.64 |
| YTX | 1043.38[b] | [d]10.53 (—) | 669 | 40.00 | 48.23 | 45.00 | 57.00 | FC | −90.19 | −86.23 | −87.68 | −82.80 |
| KRI | n.a. | n.a. | n.a. | 40.00 | 44.00 | 45.00 | 50.00 | SDWBA | −82.35 | −80.36 | −79.90 | −77.79 |

$\rho_f$—mean fish density derived from laboratory measurements with the exception of [a]which use the mean density of gas-bearing fish or [b]mean density of non-gas fish. LWR—length to width ratio taken from laboratory measurements, [c]digital images or [d]estimated mean of all other taxa; (n) Number of individual fish contributing to LWR calculations.

### (vi) Additional target strength model metrics

Mean values for sound speed in seawater (1465.836 ms$^{-1}$) and seawater density (1.0274 g ml$^{-1}$), were used in TS models, which we obtained from CTD data collected concurrently with tissue density measurements. A constant sound speed in fish tissue (1510 ms$^{-1}$) was applied for all taxa based on measured sound speed for the myctophid *Stenobrachius leucopsarus* at 4°C [31]. We did not include any resonance effect in the biomass calculations as the estimated gas radius of the species in this study was greater than 1 mm (see electronic supplementary material, figure S1), and the myctophid fish community of the Scotia Sea is dominated by adult expatriates, so juvenile fish are less likely to contribute to the signal [32]. In addition, the process of predicting NASC would have smoothed resonance peaks as multiple years of data were used to train the GAMM and predict NASC.

## (c) Predicting acoustic backscatter - GAMM

A GAMM was developed using log$_e$ transformed NASC and environmental correlates, to predict NASC for the Southern Ocean. Using R package 'mgcv' [33], scaled t family GAMMs were fitted using a restricted maximum likelihood (REML), and penalized thin plate regression splines used on all smooth terms with a conservative value of $k = 3$ to constrain overfitting. Moran's I was used to test for spatial autocorrelation on model residuals using R package 'ape' [34], and an autoregressive correlation structure of order 1 (corAR1) was subsequently specified in the GAMM [35]. From 10 candidate environmental predictors of NASC (electronic supplementary material, table S7), variance inflation factors, adjusted $R^2$, ΔAIC and BIC, were used to identify a parsimonious final GAMM with smoothing terms for net primary productivity, geostrophic current speed, sea surface temperature (SST), daylight hours and maximum percentage sea ice (electronic supplementary material, table S8). See electronic supplementary material for full GAMM fitting and specification.

The GAMM was used to predict log$_e$ NASC for the Southern Ocean in 0.25° lat-lon grid cells (figures 1b and 2b), from environmental climatology data spanning the period of 2005–2017. Mean SST [36], geostrophic current speed [37] and daylight hours [38] covered the period from October–April. Mean primary productivity [39] covered January–March as the high latitude satellite data are limited to summer months. Mean sea ice concentration [40] covered the month of September when peak sea ice occurs. Log$_e$ NASC was back-transformed into the linear domain prior to abundance calculations. Data processing flow, from net and environmental data to biomass estimation, is summarized in electronic supplementary material, figure S5.

## (d) Acoustics to fish biomass

An acoustic estimate of fish abundance ($\rho_{a_i}$) in each 0.25° grid cell was calculated from predicted NASC using equations (2.1)–(2.3) [41,42]. Relative abundance of each species ($N_i$) was first assigned to each 0.25° grid cell, by extracting the SST at each RMT25 net sample location from the SST climatology used to predict NASC (figure 1a). Net samples were allocated into one of seven 1°C SST groups (range −1°C to 6°C), and mean abundance of each species was calculated for that group (electronic supplementary material, table S9). Each 0.25° cell was then assigned a proportional fish community composition based on the grid cell SST (figure 2a). The analysis was restricted to areas with a mean SST range of −1°C to 6°C, based on the availability of net samples. TS of each species $i$ was converted to the

linear domain (backscattering cross-section $\sigma_{bs}$):

$$\sigma_{bs_i} = 10^{TS_i/10}. \tag{2.1}$$

The proportion of each species' contribution to backscatter $P_i$ is calculated by dividing the total linear backscatter of each species ($N_i \times \sigma_{bs_i}$) by the sum of linear backscatter of all species:

$$P_i = \frac{N_i \times \sigma_{bs_i}}{\sum_{i=1}^{n} N_i \times \sigma_{bs_i}}. \tag{2.2}$$

Finally, the abundance ($\rho_a$ ind. m$^{-2}$) of species $i$ is obtained from NASC (m$^2$ nmi$^{-2}$) using:

$$\rho_{a_i} = \frac{NASC \times P_i}{\sigma_{bs_i} \times 4 \times \pi \times 1852^2}. \tag{2.3}$$

Abundance in each 0.25° grid cell was converted to biomass (g m$^{-2}$) using length–weight regressions. Total fish biomass was obtained by multiplying biomass (g m$^{-2}$) by the area of each 0.25° grid cell, and summing the biomass of all fish taxa for the Scotia Sea and Southern Ocean regions.

A sensitivity analysis was conducted into the effect of altering species length (and hence TS) to estimate mean fish biomass and quantify variability in our estimates. We ran 2000 random permutations without replacement, varying each fish TS between its median, 25th and 75th percentiles of TS. As mesopelagic fish may not be responsible for all of the acoustic backscatter, we ran additional random permutations (2000 per scenario) incorporating Antarctic krill into the model at final krill densities of 32, 64 and 128 krill m$^{-2}$ based on literature [43]. Krill length frequencies from cruises JR161, JR177 and JR200 were extracted from *Krillbase* [44], and TS was calculated using the stochastic distorted wave borne approximation (SDWBA) TS model, parameterized for orientation, speed of sound and density contrast [26–28]. We also used equations in [45], to confirm that annual primary production in the region could support the levels of fish biomass estimated by our model. See electronic supplementary material for sensitivity analyses details and results.

## 3. Results

### (a) Fish target strength estimates

The non-gas bearing taxa included the myctophids *G. braueri*, *G. fraseri*, *G. nicholsi*, large individuals of *E. antarctica* (SL ≥ 51.378 mm), and the non-myctophid *Bathylagus* spp., *Notolepis* spp. and *Cyclothone* spp. [16], which were modelled using the fixed finite cylinder model [22]. The median TS ranged from −87.68 dB re 1 m$^2$ for the small, elongated *Cyclothone* spp. (median SL of 45 mm) to −68.33 dB re 1 m$^2$ for *G. nicholsi* (median SL of 137 mm), the largest species in our study (table 1). Although gas-bearing taxa were typically smaller than non-gas bearing taxa, their median TS (estimated at 500 m depth) was considerably higher, ranging from −58.11 dB re 1 m$^2$ for *K. anderssoni* to −49.06 dB re 1 m$^2$ for *E. carlsbergi* (table 1).

### (b) Acoustic biomass estimates

#### (i) Regional fish biomass estimates

The GAMM explained 57.9% of model variance, with predicted NASC showing a ubiquitous decline in colder polar waters (figure 1b). We estimate mean mesopelagic fish biomass for the Scotia Sea (defined here as the Commission for the Conservation of Antarctic Marine Living Resources; CCAMLR subareas 48.2 and 48.3; figure 1a) to be in the region 28.69 Mt (interquartile ranges (IR) = 15.67–41.10 Mt,

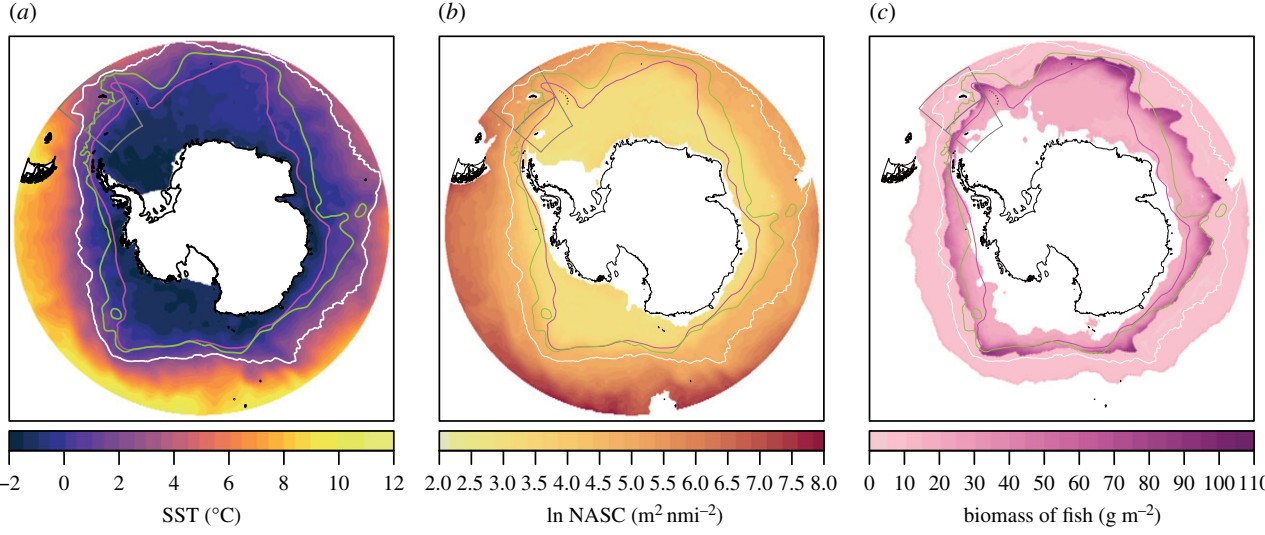

**Figure 2.** (a) Sea surface temperature climatology (October–April 2005–2017) used to predict NASC, (b) predicted $\log_e$ NASC, (c) predicted mesopelagic fish biomass for the Southern Ocean based on median SL fish, assuming that mesopelagic fish are responsible for all of the acoustic backscatter. Scotia Sea CCAMLR sub areas 48.2 and 48.2 are outlined in grey. Circumpolar lines (north to south) indicate mean positions of Antarctic Polar Front (white), Southern Antarctic Circumpolar Current Front (green) and Southern Boundary of Antarctic Circumpolar Current (magenta). (Online version in colour.)

standard deviation (s.d.) = 13.07, area = $1.57 \times 10^6$ km$^2$). The numerically abundant Myctophidae dominate this with a mean biomass of 20.60 Mt (IR = 12.11–28.14 Mt, s.d. = 8.48). For the wider Southern Ocean (figure 2, area modelled $29.51 \times 10^6$ km$^2$), mean biomass was 569.58 Mt (IR = 310.96–819.48 Mt, s.d. = 259.15), with a mean Southern Ocean myctophid biomass of 398.28 Mt (IR = 234.98–546.20, s.d. = 162.49, table 2).

### (ii) Impact of additional fauna

As other fauna can contribute to acoustic backscatter we added krill into the model. Where it is assumed that krill contribute to the acoustic signal at a relatively high abundance of 64 krill m$^{-2}$ [43], the mean biomass of mesopelagic fish was reduced to 16.96 Mt and 273.78 Mt in the Scotia Sea and Southern Ocean, respectively. This results in conservative regional myctophid biomass estimates of 12.66 Mt for the Scotia Sea and 201.04 Mt for the Southern Ocean.

### (iii) Spatial differences in biomass

The model predicted notable peaks in biomass broadly tracing the path of the Southern Antarctic Circumpolar Current Front and Southern Boundary (figure 2c). These biomass peaks are prominent towards the seasonal ice-edge (electronic supplementary material, figure S6), in regions where SST is low (less than or equal to 1°C), and dominated by non-gas bearing species. Although warmer northern latitudes have a relatively high acoustic backscatter, the fish community is dominated by highly reflective gas-bearing fish, so an overall small fish biomass results in the relatively high values of NASC (figure 1c–e). By contrast, cooler southern latitudes are characterized by gas-free fish species, so a large fish biomass is required to generate the low levels of NASC observed (figure 1c–e).

## 4. Discussion

Active acoustics indicate that net-based mesopelagic fish biomass estimates are likely to be underestimates [13]. However, interpreting acoustic data requires knowledge of the acoustic scattering properties of the local community [17]. Here, we used locally derived measurements of species acoustic properties, to develop the first species-specific TS estimates for 11 of the most common mesopelagic fish taxa from the Scotia Sea. By combining these unique TS values and relative species abundance to interpret modelled acoustic data, we have derived a Southern Ocean wide acoustic biomass estimate, which indicates that there is likely to be considerably higher fish biomass than previous net-based estimates. In addition, there are notable spatial differences that suggest a potentially larger biomass of fish in colder polar waters that previous acoustic analyses [15] are likely to have overlooked.

### (a) Biomass at the basin and oceanic scale

Within the Scotia Sea, net-based estimates of mean myctophid species biomass stands at 2.23 g m$^{-2}$ [46], when scaled to the same area as our biomass estimate this equates to a total biomass of 3.49 Mt. Using a combination of acoustic modelling and net derived community composition data, we estimate that Scotia Sea myctophid biomass to be between 12.66–20.60 Mt, 3.6–6 times larger. A previous mesopelagic fish biomass estimate for the entire Southern Ocean, when scaled to the same area ($29.5 \times 10^6$ km$^2$) as our study is 148 Mt [10]. Our mean acoustic estimate of 274 Mt if krill contribute to acoustic backscatter, or 570 Mt assuming fish are responsible for all backscatter, is approximately 1.8–3.8 times higher.

Several studies have suggested that net-based estimates are likely to be considerably lower than actual biomass. Irigoien et al. [13] proposed that global mesopelagic fish biomass was likely to be underestimated by at least an order of magnitude, while Anderson et al. [47], suggested that estimates were likely to be underestimated by a factor of approximately 2.4, similar to our estimate when krill is incorporated into the model. It is worth noting that Irigoien et al. [13], applied a single median value of TS for mesopelagic fish globally, whereas we have used locally derived TS estimates for each of the key fish taxa. It has been

**Table 2.** Total regional mesopelagic fish (all fish) and myctophid (mycto) biomass estimates in million tonnes (Mt). Biomass values calculated by varying target strength (TS) as a function of fish standard length at the median, 25th and 75th percentiles, based on 2000 random permutations of TS among each of the 11 fish taxa. 'Fish only' assumes fish are responsible for all backscatter. 'Fish (exc. krill)' are the biomass estimates when backscatter from krill was accounted for and excluded at a rate of 64 krill m$^{-2}$.

| | | | total fish biomass (Mt) | | | | | | | |
| | | | fish only | | | | fish (exc. krill) | | | |
| taxa | region | area (km$^2$) | $P_{25\%}$ | mean | $P_{75\%}$ | s.d. | $P_{25\%}$ | mean | $P_{75\%}$ | s.d. |
|---|---|---|---|---|---|---|---|---|---|---|
| all fish | Scotia Sea | 1 567 423 | 15.67 | 28.69 | 41.10 | 13.07 | 9.32 | 16.96 | 24.16 | 7.67 |
| | Southern Ocean | 29 515 433 | 310.96 | 569.58 | 819.48 | 259.71 | 149.94 | 273.78 | 389.63 | 123.94 |
| mycto | Scotia Sea | 1 567 423 | 12.11 | 20.60 | 28.14 | 8.48 | 7.42 | 12.66 | 17.36 | 5.26 |
| | Southern Ocean | 29 515 433 | 234.98 | 398.28 | 546.20 | 162.49 | 118.22 | 201.04 | 274.25 | 83.00 |

demonstrated that applying locally derived TS values from fish sampled off the Atlantic continental shelf, USA, would result in 90% fewer fish than Irigoien's estimate [13,48]. This discrepancy arises as TS is on a logarithmic scale, with each 3 dB increase resulting in a doubling of acoustic signal. Considering the median TS values in our study, the TS of the relatively small gas-bearing myctophid *K. anderssoni* is 10.22 dB higher than that of the relatively large non-gas myctophid *G. nicholsi* resulting in a return signal that is an order of magnitude higher for the considerably smaller fish. Therefore, large weakly scattering species may contribute significantly to biomass, but will be overlooked if differing scattering properties within the community are not accounted for. As TS is on a logarithmic scale, our results indicate a high level of variance; however, this is to be expected as all acoustic biomass calculations involve the conversion of data from log to linear scales.

It has previously been reported that there is a latitudinal switch in the mesopelagic fish community, from taxa with highly reflective swimbladders at warmer latitudes, to larger non-gas bearing taxa closer to the Antarctic continent [16]. Applying our taxa specific TS estimates across the Southern Ocean, demonstrates how changing communities can result in distinctly different scattering regimes. A notable effect of the observed latitudinal change in the community [16], is a predicted peak in biomass in the region between the ice-edge and SST $\leq 1°$C, despite an overall decline in acoustic backscatter at higher latitudes (figure 2*b,c*). Similar levels of mesopelagic fish biomass have been estimated in the western Pacific sector of the Southern Ocean region, by applying TS values from ecologically similar species [49]. At the global scale, the Southern Ocean has predominantly been viewed as having a relatively low mesopelagic fish biomass [15], with higher densities found in productive tropical and subtropical regions [1]. While our fish biomass estimate is relatively high in comparison to global net-based estimates of 1000 Mt [1], a simple calculation [45] indicates that an annual primary production of 54 g C m$^{-2}$ a$^{-1}$ [50] is sufficient to sustain the requirements of such high levels of fish and krill at 29.4 g C m$^{-2}$ a$^{-1}$. However, reliable comparisons between acoustic data from polar and temperate regions clearly requires additional information on latitudinal changes in community composition and morphology at the global scale.

## (b) Methodological considerations

Demer [51] summarized the causes of uncertainty that are liable to produce errors in acoustic biomass estimates, considering data collection, calibration, processing and environmental conditions. The largest sources of error are likely to derive from species identification, TS measurements and fish vertical migration [51]. We considered each of these issues to limit uncertainty in our biomass estimates as follows.

We resolved species identification by using Scotia Sea net sample data to locate scattering communities by common SST values. While direct observation of community structure across the Southern Ocean would be ideal, our approach seems biologically reasonable as the species in our study have a circumpolar distribution [52,53], with thermal niche and latitudinal preference the dominant structuring factors in these communities [54]. Notably, a recent study from the western Pacific sector of the Southern Ocean found broadly similar taxa [49]. We also acknowledge that net sampling methods have an inherent degree of bias and may miss strongly scattering taxa such as gas-bearing siphonophores that are poorly sampled by nets [42,55] or favour the capture of slower individuals [56]. Net bias may result in skewed community composition, affecting the relative proportions of gas and non-gas bearing taxa, resulting in either an over- or under-estimation of fish biomass as the cryptic portion of the community is unaccounted for. A predominance of large non-gas-bearing taxa in our net samples would likely lead to an overestimation of biomass, and so these initial biomass estimates should be treated with caution. However, nets are currently the best available method for identifying community composition. While still an emerging technique, analysis of environmental DNA from seawater samples holds potential to resolve issues of community composition and relative abundance of taxa, including identifying the presence of cryptic species that are poorly sampled or destroyed by nets [57,58]. While net sample data are required to validate size spectra, environmental DNA could signal changes to community structure [59], allowing for adjustments to future model predictions.

Estimates of TS would ideally be derived empirically by insonifying live fish behaving naturally [18]. However, the collection of such data is challenging for mesopelagic fish as cameras lack the resolution to identify taxa to species

level [55], and measurement at the surface requires intact fish that are equilibrated to atmospheric pressure, rarely achievable in fish sampled from mesopelagic depths. Given these challenges we used established models to derive taxa specific TS estimates based on empirical measurements of fish physiology and density. During vertical migration changes in swimbladder gas volume or pressure can result in resonance (leading to disproportionately high backscatter), which is particularly problematic in small gas-bearing species at lower frequencies [18,60]. We suggest that fish migration is likely to have had only limited impact in our study, as the GAMM was trained with both day and night data and the climatologies used to predict NASC (SST, primary productivity, geostrophic current speed, daylight hours and sea ice concentration) were averaged over multiple years and seasons.

Our model predicts a peak in biomass north of the seasonal ice-edge but within the cooler southern regions of the Antarctic Circumpolar Current, that is driven by the shift in community composition at colder temperatures, as evidenced from our net sample data. While this peak in biomass generally tracks mean frontal positions, there is a dissociation in the Indian Ocean and eastern Pacific sectors of the Southern Ocean (figure 2c). Conceptually, fronts delineate water masses, and hence habitats, with distinct physical properties that are likely to structure mesopelagic communities residing at depth [52,53]. However, frontal features are highly dynamic and large-scale sub-surface oceanographic measurements are challenging to obtain, with the Southern Ocean being particularly data poor [61]. In addition, there has been an increase in surface warming in the northern regions of the Antarctic Circumpolar Current, with a concurrent cooling in southern surface waters [62] that may amplify the effects of changing community composition on biomass estimates in these poorly sampled regions. We used satellite-derived climatologies of surface variables including primary productivity, daylight hours, geostrophic current speed, sea ice extent and SST, to train our GAMM that accounted for over 57.9% of the variability in our measured NASC data. We also considered the use of geopotential height as a proxy for water masses and modelled temperature at 200 m; however, SST had a stronger correlation with acoustic backscatter and also had the benefit of being readily available as a satellite product. Seasonal sea ice extent is also likely to be important to the spatial structuring of the fish in our study, as while some are polar specialists, none are known to be cryopelagic. Refinement of our biomass estimates will require additional acoustic and community composition data for under-sampled regions of the Indian Ocean and eastern Pacific, alongside an improved understanding of the environmental variables that determine the structure of the fish assemblages.

## (c) Implications for the ecosystem

The predicted peak in mesopelagic fish biomass towards the seasonal ice-edge has relevance to our understanding of the open ocean foraging behaviour of land-breeding predators in the polar frontal zone. King penguins specialize on myctophids and have been tracked foraging in the region between the Antarctic Polar Front and Antarctic Circumpolar Current Front, presumably to access rich reliable food resources [63–

65]. Specifically, tracking data reveal that when king penguin chicks are in the crèche stage, adults forage up to 1600 km or more, reaching latitudes where this study predicts biomass peaks [65,66]. While there are no existing data on the species they are consuming at these high latitudes, it seems likely that these would be the biomass dominant *E. antarctica* [66]. The ice-edge and Weddell–Scotia Confluence, are also known to be important foraging grounds for number of seabirds, with *E. antarctica* being a main prey item [67]. Elephant seals also exploit this potentially high biomass region throughout the year, with king and macaroni penguins, elephant and fur seals all foraging towards the ice-edge in winter [68].

While the Southern Ocean's top predators may currently be able to exploit a sizeable biomass of fish, this resource is at risk. It has been predicted that future ocean-warming will drive a poleward shift in mesopelagic fish species [54]. This change in community is likely to result in a reduction in fish biomass as increasing temperatures have been shown to result in reductions in fish body size [69], and Southern Ocean myctophids comply with Bergman's rule, where smaller taxa occupy warmer waters [70]. Any significant reduction in fish biomass is likely to have profound implications for Antarctic food web dynamics, as higher predators encounter smaller prey, but also as larger zooplankton are released from predation by larger fish. There are also likely to be consequences for the biological carbon pump and energy transfer to deeper oceanic layers, as larger mesopelagic fish are predicted to sequester more carbon than smaller taxa through diel vertical migration [2]. We propose that monitoring of communities via acoustics may provide a mechanism for the early detection of structural change within the mesopelagic fish community. While we currently lack sufficient time series to monitor this change, the increasing use of ships of opportunity [49] and autonomous systems [71] to collect calibrated acoustic data could help build the time-series required and fill knowledge gaps in data poor regions.

Data accessibility. All data are publicly available as part of the electronic supplementary material [72] and from the UK Polar Data Centre [73].

Authors' contributions. T.D.: conceptualization, data curation, formal analysis, investigation, methodology, project administration, software, validation, visualization, writing—original draft, writing—review and editing; S.F.: conceptualization, funding acquisition, investigation, methodology, resources, supervision, writing—review and editing; R.A.S.: data curation, investigation, supervision, writing—review and editing; M.J.G.: funding acquisition, investigation, methodology, supervision, writing—review and editing.

All authors gave final approval for publication and agreed to be held accountable for the work performed therein.

Competing interests. We declare we have no competing interests.

Funding. The scientific cruises (the Polar Ocean Ecosystem Time Series—Western Core Box), T.D., R.A.S. and S.F. are funded as part of the Ecosystems Programme at the British Antarctic Survey, Natural Environment Research Council, a part of UK Research and Innovation. T.D. was also supported by a NERC GW4+ Doctoral Training Partnership award NE/L002434/1. Stewardship of the acoustic data received support from the European H2020 International Cooperation project MESOPP (Mesopelagic Southern Ocean Prey and Predators, www.mesopp.eu).

Acknowledgements. We thank the crew and scientists of the *RRS James Clark Ross* for support in sample and acoustic data collection, and Tim Stanton for helpful discussions on acoustic scattering models.

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
