## [Peer Review File · Proceedings of the Royal Society B: Biological Sciences]

Review History

RSPB-2021-1781.R0 (Original submission)

Review form: Reviewer 1

Recommendation

Major revision is needed (please make suggestions in comments)

Scientific importance: Is the manuscript an original and important contribution to its field?

Good

General interest: Is the paper of sufficient general interest?

Good

Quality of the paper: Is the overall quality of the paper suitable?

Acceptable

Is the length of the paper justified?

Yes

Should the paper be seen by a specialist statistical reviewer?

No

Do you have any concerns about statistical analyses in this paper? If so, please specify them explicitly in your report.

Yes

It is a condition of publication that authors make their supporting data, code and materials available - either as supplementary material or hosted in an external repository. Please rate, if applicable, the supporting data on the following criteria.

Is it accessible?

N/A

Is it clear?

N/A

Is it adequate?

N/A

Do you have any ethical concerns with this paper?

No

Comments to the Author

Overall the paper is well written. One observation is that the supplementary material is more concise and easier to read compared to the main methods section. The authors should consider bringing some of the material in the SI into the methods. The length should be about the same, but it needs to be more accurate.

Net sample distribution is used to directly distribute backscatter among the species (Line 162). This has a few challenges. It relies on catchability being similar between all taxonomic groups contributing to the NASC, that the sampling covers the same vertical extent as the NASC values, and that the acoustic sampling is unbiased. Net sampling are selective (and typically size dependent), and this may cause biases in the proportions, which will again lead to biases in the overall abundance. This is particularly important since the TS varies substantially between species with and without swim bladder. The net sampling described in line 72 and onwards does not provide any insight to this. The authors uses sensitivity analysis to get a handle on how much the estimates are influenced by this, which is a good strategy to address this point. But I lack some discussions about the problem. The sensitivity analysis also need some attention, see comment below.

Eq S19 is not using conventional notation. Typically σ_{bs} is reserved for backscatter from one scatterer. I would recommend to not use $\sigma_{bs_totalfish}$ in Eq 19. To avoid this perhaps just use the σ_{bs} weighted proportion directly, for example $p_i = (N_i \cdot \sigma_{bs_i}) / (\sum_i N_i \cdot \sigma_{bs_i})$, where p_i is the proportion. Then the following density estimate is straight forward (Eq. 21). This leads to another question. From reviewing the equations, the proportions sum up to one, but what is then the unaccounted backscatter in Figure S6? Are the N for fish readjusted after assigning some backscatter to Krill? The algorithm seem very ad hoc. Why not include Krill NASC in the denominator of Eq S20? This will be a cleaner implementation. The input to the "model" will then be the catches of fish (N_i) as well as an assumed number and σ_{bs} of krill.

Another cause of bias may be caused by non-fish scatterers, like siphonophores. These are typically not retained in the nets and can bias the results substantially. I am not familiar enough with the ecosystem and do not know whether smaller gas-bearing organisms are present or not, but the authors need to at least confirm the absence of other gas-bearing organisms in the area or not. If not, the abundance of such organisms could be included similar to how the Krill has been included.

With the suggested approach above one set of equations can be used for all the steps, making the ad hoc approach presented in Figure S6 unnecessary. Another benefit from this approach is that the standard sensitivity analysis could be employed. The approach used is in reality a local sensitivity analysis where one variable is varied while keeping the rest constant. This works for assessing the sensitivity at one location in the parameter space, hence “local”. There are metrics that can be calculated for sensitivity analysis that should be used. The authors should also consider other (global) sensitivity methods to get a handle on the uncertainty. In any case, an established sensitivity metric is a requirement, as opposed to the ad hoc approach reported in Table S10. In addition, the sensitivity analysis does not address the sensitivity in the parameters for TS or for the TS models itself, probably underestimating the variance estimate of the abundance.

The authors (in the abstract) claims that the TS of dominant mesopelagic fish in the Scotia Sea is estimated for the first time. The authors have used a simplified model (finite cylinder) for non-swimbladdered fish and have not validated their estimated TS against any field measurements. Although the approach is appropriate for the methods used in the paper, they cannot claim that they have provided a reliable TS model/estimate for the mesopelagic fish. The same concerns exist for swimbladdered mesopelagic fish.

The authors claim that they “...did not include any resonance effect in the TS models...” (Line 129). The models described in equations S11-14 are resonant backscattering models even if the resonance frequency occur lower than 38 kHz. For several smaller size groups, especially at deeper waters, the resonant scatterers dominates the backscatter. Authors need to provide evidence that smaller gas-bearing organisms are properly sampled, or, if not, include that in the sensitivity analysis.

Review form: Reviewer 2

Recommendation

Major revision is needed (please make suggestions in comments)

Scientific importance: Is the manuscript an original and important contribution to its field?

Good

General interest: Is the paper of sufficient general interest?

Good

Quality of the paper: Is the overall quality of the paper suitable?

Good

Is the length of the paper justified?

Yes

Should the paper be seen by a specialist statistical reviewer?

No

Do you have any concerns about statistical analyses in this paper? If so, please specify them explicitly in your report.

Yes

It is a condition of publication that authors make their supporting data, code and materials available - either as supplementary material or hosted in an external repository. Please rate, if applicable, the supporting data on the following criteria.

Is it accessible?

N/A

Is it clear?

N/A

Is it adequate?

N/A

Do you have any ethical concerns with this paper?

No

Comments to the Author

Review of Dornan et al. "Large mesopelagic fish biomass in the Southern Ocean resolved by acoustic properties", RSPB-2021-1781.

The authors report on an approach to estimate the cryptic biomass of mesopelagic fish in the Southern Ocean. By using some generic, but Antarctic specific, estimates of acoustic properties and basic shapes of dominant mesopelagic fishes, compositional data from some net tows collected in several areas of the Atlantic Sector of the Southern Ocean, acoustic data (38 kHz) taken along transects over a six year period, and climatological satellite-based data, the authors estimate that krill biomass in the Southern Ocean is between 300 and 700 million tonnes, or up to 70% of the most recent global estimate. The paper is well-written and clear. The authors do a good job of describing the caveats of the work they have produced. However, uncertainty (rather than calculated range) of estimates is really not well described. More thought and specific need to be incorporated within the discussion (see below).

This paper needs some additional specificity in the discussion, and as such should be considered for publication after considering the following comments.

Discussion

Line 270-274

lines to 272-274. While it is true that the TS scaling the biomass is probably the single biggest issue for uncertainty in the definition of an acoustic model for the different species, and the proper weighting of mesopelagic fish biomass in a sampled area, uncertainty regarding the spatial model used to extrapolate the data from the Atlantic to the remainder of the Southern Ocean using the GAMM is probably a bigger over uncertainty to the total biomass. See Figure 2. Although the authors argue that the model is a good fit, it really is not well constrained, and there is no evidence that the surface features (seasonal SST) are really properly describing the subsurface features that define the boundaries of different communities around the Southern Ocean. For example there are areas where seasonal average SST does not match the position of fronts as defined by Orsi (and visualized in this paper). As the seasonal SST and the Orsi fronts describe different dynamical aspects of the ocean, the difference between the two and its implications on estimating biomass are important.

If the 700 million tonne value is to be believable, the authors should provide some calculations that the system can actually support an equal biomass of Antarctic krill and of mesopelagic fish.

Line 275 to 288

Next lines from 275 to 288 the authors quite nicely note the differences in the selectivity and availability of different species to the nets that have been used. With a larger number of nets tows the probability of missing taxa could be estimated, but in this case it is likely that the compositional changes and dominance are probably the biggest issues, rather than the possibility of missing some species. This the allocation of energy among species with similar (+/- 2dB) TS might be a bigger issue. Of course the total lack of samples from the Pacific is a large uncertainty, as noted. The hypothesis that community and biomass density of an entire ocean basin that is >

50% of the Southern Ocean is similar enough that the SST can provide an adequate constraint is difficult to understand. The Scotia Sea and Southwest Atlantic has the highest primary production of the Southern Ocean, and likely can support more biomass. What is the difference in mean summer primary production in these areas... could the biomass be constrained by the differences in primary production.

Lines 297 - 298 be more specific about what the climatology is.. Is this SST?

Line 300 to 311 this first paragraph is rather specific to South Georgia and to King penguins in that area. Are the spatial patterns likely to really mostly effect king penguins... The location of production around the Antarctic in these fronts could be extremely important for overwinter migrations by chinstrap penguins that have been observed to migrate from the Peninsula to nearly the Ross Sea to the west along the ice edge. It seems like some taxa might be aided by the southern movement of the fronts in the future, although, the authors have not indicated how far these features are likely to move (e.g. 50km average, 100 km, 250km) and this makes difficult to evaluate the conclusions that such changes will result in profound effects (line 317).

Line 312-322 the authors are rather superficial in the discussion of the impacts to mesopelagic fish resources. What are the likely changes in sub-surface temperatures in the vast majority of the system, how much has it changed in the last 50 years, if projected to the next 50, would the species response be observable?

The argument that the acoustic signal may provide an early indication of structural change is interesting. However what is the current inter-annual or inter-seasonal variability in the acoustic signals in different regions. How long a time series would be required to see a 10% decline in TS that would result in a defined percentage change in biomass. Would this be the same farther south from the principal sampling locations. Would this be the same in the Pacific where primary production is also likely to change with warming and climate change

I would challenge the authors to go further to constrain the effects. Otherwise the results of this paper will either be cited as a major result, or derided as too speculative.

Figure 2

Figure 2. The high biomass of fish predicted by the model is interesting, However in Figure 2 there seems to be a large discrepancy between the location of the fronts that drive the biomass and the seasonal SS. This is especially noticeable in the Indian Ocean sector where the high biomass is associated with SST gradient, but is well north of the position of the Southern Antarctic Circumpolar Current Front. This disconnect also occurs in the Southeast Pacific near the West Antarctic peninsula. So it would be interesting to understand these difference impact the likely outcomes of the model. To this reviewer, these differences suggest a disconnect between the oceanography that generates the subsurface features defining the spatial habitat, and the surface expression of a features that are weakly correlated with the dynamics in some areas. The authors should consider what impact this could have on their estimates.

Decision letter (RSPB-2021-1781.R0)

06-Oct-2021

Dear Dr Dornan:

Your manuscript has now been peer reviewed and the reviews have been assessed by an Associate Editor. The reviewers' comments (not including confidential comments to the Editor) and the comments from the Associate Editor are included at the end of this email for your reference. As you will see, the reviewers and the Editors have raised some concerns with your manuscript and we would like to invite you to revise your manuscript to address them.

Research ethics:

Use of animals and field studies:

It is a condition of publication that you make available the data and research materials supporting the results in the article. Please see our Data Sharing Policies (<https://royalsociety.org/journals/authors/author-guidelines/#data>). Datasets should be deposited in an appropriate publicly available repository and details of the associated accession number, link or DOI to the datasets must be included in the Data Accessibility section of the article (<https://royalsociety.org/journals/ethics-policies/data-sharing-mining/>). Reference(s) to datasets should also be included in the reference list of the article with DOIs (where available).

Please submit a copy of your revised paper within three weeks. If we do not hear from you within this time your manuscript will be rejected. If you are unable to meet this deadline please let us know as soon as possible, as we may be able to grant a short extension.

Best wishes,
Dr Locke Rowe
mailto:proceedingsb@royalsociety.org

Associate Editor

Board Member: 1

Comments to Author:

Both reviewers have identified a number of technical issues with this manuscript as it currently stands, primarily in regards to the modelling and algorithms used, assessing bias associated with species not sampled and a lack of discussion around associated uncertainties. Both reviewers have provided constructive comments on how these issues could be addressed. Please ensure that revisions directly respond to suggestions provided.

Reviewer(s)' Comments to Author:

Referee: 1

Comments to the Author(s)

Overall the paper is well written. One observation is that the supplementary material is more concise and easier to read compared to the main methods section. The authors should consider bringing some of the material in the SI into the methods. The length should be about the same, but it needs to be more accurate.

Net sample distribution is used to directly distribute backscatter among the species (Line 162). This has a few challenges. It relies on catchability being similar between all taxonomic groups contributing to the NASC, that the sampling covers the same vertical extent as the NASC values, and that the acoustic sampling is unbiased. Net sampling are selective (and typically size dependent), and this may cause biases in the proportions, which will again lead to biases in the overall abundance. This is particularly important since the TS varies substantially between species with and without swim bladder. The net sampling described in line 72 and onwards does not provide any insight to this. The authors uses sensitivity analysis to get a handle on how much the estimates are influenced by this, which is a good strategy to address this point. But I lack some discussions about the problem. The sensitivity analysis also need some attention, see comment below.

Eq S19 is not using conventional notation. Typically σ_{bs} is reserved for backscatter from one scatterer. I would recommend to not use $\sigma_{bs_totalfish}$ in Eq 19. To avoid this perhaps just use the σ_{bs} weighted proportion directly, for example $p_i = (N_i \cdot \sigma_{bs_i}) / (\sum_i N_i \cdot \sigma_{bs_i})$, where p_i is the proportion. Then the following density estimate is straight forward (Eq. 21). This leads to another question. From reviewing the equations, the proportions sum up to one, but what is then the unaccounted backscatter in Figure S6? Are the N for fish readjusted after assigning some backscatter to Krill? The algorithm seem very ad hoc. Why not include Krill NASC in the denominator of Eq S20? This will be a cleaner implementation. The input to the “model” will then be the catches of fish (N_i) as well as an assumed number and σ_{bs} of krill.

Another cause of bias may be caused by non-fish scatterers, like siphonophores. These are typically not retained in the nets and can bias the results substantially. I am not familiar enough with the ecosystem and do not know whether smaller gas-bearing organisms are present or not, but the authors need to at least confirm the absence of other gas-bearing organisms in the area or not. If not, the abundance of such organisms could be included similar to how the Krill has been included.

With the suggested approach above one set of equations can be used for all the steps, making the ad hoc approach presented in Figure S6 unnecessary. Another benefit from this approach is that the standard sensitivity analysis could be employed. The approach used is in reality a local sensitivity analysis where one variable is varied while keeping the rest constant. This works for assessing the sensitivity at one location in the parameter space, hence “local”. There are metrics that can be calculated for sensitivity analysis that should be used. The authors should also consider other (global) sensitivity methods to get a handle on the uncertainty. In any case, an established sensitivity metric is a requirement, as opposed to the ad hoc approach reported in Table S10. In addition, the sensitivity analysis does not address the sensitivity in the parameters for TS or for the TS models itself, probably underestimating the variance estimate of the abundance.

The authors (in the abstract) claims that the TS of dominant mesopelagic fish in the Scotia Sea is estimated for the first time. The authors have used a simplified model (finite cylinder) for non-swimbladdered fish and have not validated their estimated TS against any field measurements. Although the approach is appropriate for the methods used in the paper, they cannot claim that they have provided a reliable TS model/estimate for the mesopelagic fish. The same concerns exist for swimbladdered mesopelagic fish.

The authors claim that they “...did not include any resonance effect in the TS models...” (Line 129). The models described in equations S11-14 are resonant backscattering models even if the resonance frequency occur lower than 38 kHz. For several smaller size groups, especially at deeper waters, the resonant scatterers dominates the backscatter. Authors need to provide evidence that smaller gas-bearing organisms are properly sampled, or, if not, include that in the sensitivity analysis.

Referee: 2

Comments to the Author(s)

Review of Dornan et al. “Large mesopelagic fish biomass in the Southern Ocean resolved by acoustic properties”, RSPB-2021-1781.

The authors report on an approach to estimate the cryptic biomass of mesopelagic fish in the Southern Ocean. By using some generic, but Antarctic specific, estimates of acoustic properties and basic shapes of dominant mesopelagic fishes, compositional data from some net tows collected in several areas of the Atlantic Sector of the Southern Ocean, acoustic data (38 kHz) taken along transects over a six year period, and climatological satellite-based data, the authors

estimate that krill biomass in the Southern Ocean is between 300 and 700 million tonnes, or up to 70% of the most recent global estimate. The paper is well-written and clear. The authors do a good job of describing the caveats of the work they have produced. However, uncertainty (rather than calculated range) of estimates is really not well described. More thought and specific need to be incorporated within the discussion (see below).

This paper needs some additional specificity in the discussion, and as such should be considered for publication after considering the following comments.

Discussion

Line 270-274

lines to 272-274. While it is true that the TS scaling the biomass is probably the single biggest issue for uncertainty in the definition of an acoustic model for the different species, and the proper weighting of mesopelagic fish biomass in a sampled area, uncertainty regarding the spatial model used to extrapolate the data from the Atlantic to the remainder of the Southern Ocean using the GAMM is probably a bigger over uncertainty to the total biomass. See Figure 2.

Although the authors argue that the model is a good fit, it really is not well constrained, and there is no evidence that the surface features (seasonal SST) are really properly describing the subsurface features that define the boundaries of different communities around the Southern Ocean. For example there are areas where seasonal average SST does not match the position of fronts as defined by Orsi (and visualized in this paper). As the seasonal SST and the Orsi fronts describe different dynamical aspects of the ocean, the difference between the two and its implications on estimating biomass are important.

If the 700 million tonne value is to be believable, the authors should provide some calculations that the system can actually support an equal biomass of Antarctic krill and of mesopelagic fish.

Line 275 to 288

Next lines from 275 to 288 the authors quite nicely note the differences in the selectivity and availability of different species to the nets that have been used. With a larger number of nets tows the probability of missing taxa could be estimated, but in this case it is likely that the compositional changes and dominance are probably the biggest issues, rather than the possibility of missing some species. This the allocation of energy among species with similar (+/- 2dB) TS might be a bigger issue. Of course the total lack of samples from the Pacific is a large uncertainty, as noted. The hypothesis that community and biomass density of an entire ocean basin that is > 50% of the Southern Ocean is similar enough that the SST can provide an adequate constraint is difficult to understand. The Scotia Sea and Southwest Atlantic has the highest primary production of the Southern Ocean, and likely can support more biomass. What is the difference in mean summer primary production in these areas... could the biomass be constrained by the differences in primary production.

Lines 297 - 298 be more specific about what the climatology is.. Is this SST?

Line 300 to 311 this first paragraph is rather specific to South Georgia and to King penguins in that area. Are the spatial patterns likely to really mostly effect king penguins... The location of production around the Antarctic in these fronts could be extremely important for overwinter migrations by chinstrap penguins that have been observed to migrate from the Peninsula to nearly the Ross Sea to the west along the ice edge. It seems like some taxa might be aided by the southern movement of the fronts in the future, although, the authors have not indicated how far these features are likely to move (e.g. 50km average, 100 km, 250km) and this makes difficult to evaluate the conclusions that such changes will result in profound effects (line 317).

Line 312-322 the authors are rather superficial in the discussion of the impacts to mesopelagic fish resources. What are the likely changes in sub-surface temperatures in the vast majority of the system, how much has it changed in the last 50 years, if projected to the next 50, would the species response be observable?

The argument that the acoustic signal may provide an early indication of structural change is interesting. However what is the current inter-annual or inter-seasonal variability in the acoustic signals in different regions. How long a time series would be required to see a 10% decline in TS

that would result in a defined percentage change in biomass. Would this be the same farther south from the principal sampling locations. Would this be the same in the Pacific where primary production is also likely to change with warming and climate change
I would challenge the authors to go further to constrain the effects. Otherwise the results of this paper will either be cited as a major result, or derided as too speculative.

Figure 2

Figure 2. The high biomass of fish predicted by the model is interesting, However in Figure 2 there seems to be a large discrepancy between the location of the fronts that drive the biomass and the seasonal SS. This is especially noticeable in the Indian Ocean sector where the high biomass is associated with SST gradient, but is well north of the position of the Southern Antarctic Circumpolar Current Front. This disconnect also occurs in the Southeast Pacific near the West Antarctic peninsula. So it would be interesting to understand these difference impact the likely outcomes of the model. To this reviewer, these differences suggest a disconnect between the oceanography that generates the subsurface features defining the spatial habitat, and the surface expression of a features that are weakly correlated with the dynamics in some areas. The authors should consider what impact this could have on their estimates.

Author's Response to Decision Letter for (RSPB-2021-1781.R0)

See Appendix A.

Decision letter (RSPB-2021-1781.R1)

21-Dec-2021

Dear Dr Dornan

I am pleased to inform you that your manuscript RSPB-2021-1781.R1 entitled "Large mesopelagic fish biomass in the Southern Ocean resolved by acoustic properties" has been accepted for publication in Proceedings B.

The AE has recommended publication, but also suggest some minor revisions to your manuscript. Therefore, I invite you to respond to the AE's comments and revise your manuscript. Because the schedule for publication is very tight, it is a condition of publication that you submit the revised version of your manuscript within 7 days. If you do not think you will be able to meet this date please let us know.

Sincerely,
Dr Locke Rowe
Editor, Proceedings B
mailto:proceedingsb@royalsociety.org

Associate Editor:

Comments to Author:

The authors have largely addressed the reviewers comments in revising their manuscript. However, there are a couple of points raised by the reviewers that the authors have seemingly passed over superficially and could be more fully addressed with the addition/revision of some of the content in the discussion in identifying the uncertainties of the study, which both authors are calling to be more adequately addressed. I would consider that this would only entail a sentence or two and these minor considerations will strengthen the manuscript and better respond to the reviewers comments.

In particular, the second reviewer raised the issue of the use of surface features in describing complex subsurface features and how this might contribute to uncertainty in modelled outputs. The authors have passed over this comment by identifying that the reviewers reference to Orsi was not necessarily appropriate and referencing one publication to provide evidence that surface features adequately reflect subsurface features. Given the complexity of features associated with frontal regions, varying views on understanding of these features and how they are changing both directly as a result of increased carbon uptake and also changes in climate phenomena that influence them (see the IPCC's SROCC), I'd suggest that the authors include some small revisions to the discussion identifying some of the limitations associated with use of SST to describe complex subsurface features (particularly when studying biological features that only spend part of their day in the upper parts of the ocean). They may want to discuss this in light of the correlations between SST and SST200 and why SST might be considered more appropriate. This will strengthen the manuscripts consideration of uncertainty.

Further, the second reviewer queried the disconnect between the areas of high biomass estimated by the model and the location of fronts, with the areas of high biomass north of frontal averages. The authors have responded with a justification centered around the extent of sea ice. While I agree that sea ice extent will no doubt have some influence on distributions of species that are not associated with the cryosphere, I'm not sure I fully understand this justification given the northward location of the high biomass (and sampling outside of the maximum sea ice extent), so in regions beyond the relevant seasonal northward limits of sea ice extent. In considering oceanographic features that might influence species compositions and distributions more broadly, authors might want to consider the higher rate of ocean warming northward of the ACC and the oceanographic processes contributing to this (again see the IPCC SROCC) and what influences these changes might be having on productivity in these northern regions of the Southern Ocean rather than solely focusing on sea ice extent. As identified above this would only require minor revision to the discussion and would strengthen the discussion on uncertainty.

In addition I have attached some minor editorial revisions for consideration by the authors in finalising their manuscript.

Author's Response to Decision Letter for (RSPB-2021-1781.R1)

See Appendix B.

Decision letter (RSPB-2021-1781.R2)

23-Dec-2021

Dear Dr Dornan

I am pleased to inform you that your manuscript entitled "Large mesopelagic fish biomass in the Southern Ocean resolved by acoustic properties" has been accepted for publication in Proceedings B.

If you are likely to be away from e-mail contact please let us know. Due to rapid publication and an extremely tight schedule, if comments are not received, we may publish the paper as it stands. If you have any queries regarding the production of your final article or the publication date please contact procb_proofs@royalsociety.org

Data Accessibility section

Open Access

You are invited to opt for Open Access, making your freely available to all as soon as it is ready for publication under a CCBY licence. Our article processing charge for Open Access is £1700. Corresponding authors from member institutions (<http://royalsocietypublishing.org/site/librarians/allmembers.xhtml>) receive a 25% discount to these charges. For more information please visit <http://royalsocietypublishing.org/open-access>.

Paper charges

Sincerely,

Appendix A

Response to reviewers

(Please note that all line numbers relate to track change in simple mark up.)

Reviewer 1	Response
1.1 Overall the paper is well written. One observation is that the supplementary material is more concise and easier to read compared to the main methods section. The authors should consider bringing some of the material in the SI into the methods. The length should be about the same, but it needs to be more accurate.	We had purposely kept sections of our methodology brief as these had been published in detail in our referenced publication Dornan et al. 2010. However, we acknowledge that this may have made the methods unclear. We have taken the reviewers advice and slightly restructured the main papers methods section, including additional detail from the SI to improve clarity on our methods. We have also updated our methodology and results to include new sensitivity analyses as requested.
1.2 Net sample distribution is used to directly distribute backscatter among the species (Line 162). This has a few challenges. It relies on catchability being similar between all taxonomic groups contributing to the NASC, that the sampling covers the same vertical extent as the NASC values, and that the acoustic sampling is unbiased. Net sampling are selective (and typically size dependent), and this may cause biases in the proportions, which will again lead to biases in the overall abundance. This is particularly important since the TS varies substantially between species with and without swim bladder. The net sampling described in line 72 and onwards does not provide any insight to this. The authors uses sensitivity analysis to get a handle on how much the estimates are influenced by this, which is a good strategy to address this point. But I lack some discussions about the problem. The sensitivity analysis also need some attention, see comment below.	We agree with the reviewer and recognise the implications of inherent net sampling bias. We have now included a more detailed description of net sampling (Lines 84-86), which confirms that our net sampling covered the same vertical resolution as acoustic samples. In addition we have updated our discussion Lines 295-301, to read “We also acknowledge that net sampling methods have an inherent degree of bias, which may miss strongly scattering taxa such as gas-bearing siphonophores that are poorly sampled by nets [39, 55] or favour the capture of slower individuals [56]. Net bias may result in skewed community composition, affecting the relative proportions of gas and non-gas bearing taxa, resulting in either an over or under-estimation of fish biomass as the cryptic portion of the community is unaccounted for. A predominance of large non-gas bearing taxa in our net samples would likely lead to an overestimation of biomass, and so these initial biomass estimates should be treated with caution. ”
1.3 Eq S19 is not using conventional notation. Typically σ_{bs} is reserved for backscatter from one scatterer. I would recommend to not use $\sigma_{bs_totalfish}$ in Eq 19. To avoid this perhaps just use the σ_{bs} weighted proportion directly, for example $p_i = \frac{N_i \sigma_{bs_i}}{\sum_i N_i \sigma_{bs_i}}$, where p_i is the proportion. Then the following density estimate is straight forward (Eq. 21).	We have updated the equation annotation as suggested throughout. We have also included these equations in the main manuscript Lines 177-186, Eq. 1-3, to address the reviewer’s first comment.

1. 4 From reviewing the equations, the proportions sum up to one, but what is then the unaccounted backscatter in Figure S6? Are the N for fish readjusted after assigning some backscatter to Krill? The algorithm seem very ad hoc. Why not include Krill NASC in the denominator of Eq S20? This will be a cleaner implementation. The input to the “model” will then be the catches of fish (N_i) as well as an assumed number and sigma bs of krill.	The reviewer has highlighted a potential source of confusion, which we have addressed by updating the explanation in the supplementary text and equation annotation. The reviewer is correct when they ask if the N for fish are readjusted after assigning some backscatter to krill. Final krill density was estimated from the literature. This should have been annotated in Eq. S21 (now Eq. S19) as $\rho_a \text{ krill}$ not N_{krill} (i.e. final abundance rather than net abundance). We have now updated these equations and clarified our procedure in the supplementary text to read “Krill abundance ($\rho_a \text{ krill. m}^{-2}$) estimates were taken from literature, and applied at a value of 64 krill m^{-2} throughout the Southern Ocean [26], and their influence tested by halving and doubling this value. As final krill abundance per unit area was estimated from literature, it was necessary to reduce the proportion of backscatter attributable to fish in the model. The theoretical number of krill in a net N_{krill} was back calculated using Equations S19-S20:...”
1.5 Another cause of bias may be caused by non-fish scatterers, like siphonophores. These are typically not retained in the nets and can bias the results substantially. I am not familiar enough with the ecosystem and do not know whether smaller gas-bearing organisms are present or not, but the authors need to at least confirm the absence of other gas-bearing organisms in the area or not. If not, the abundance of such organisms could be included similar to how the Krill has been included.	We acknowledge that the presence of gas-bearing siphonophores could contribute to acoustic signal. However, there is currently a lack of data on the relative abundance of siphonophores within the Southern Ocean or their scattering properties. Proud et al. (2018) aimed to address this by classing siphonophore contribution as similar to a gas bearing fish, concluding that if 10% of the gas induced scatter came from siphonophores there would be 10% fewer gas bearing fish. We raise the issue of cryptic taxa such as siphonophores in the discussion (Lines 295-307), but have not included siphonophores in the model as calculating an appropriate portion of backscatter attributable to siphonophores would be highly speculative at this time.

1.6 With the suggested approach above one set of equations can be used for all the steps, making the ad hoc approach presented in Figure S6 unnecessary. Another benefit from this approach is that the standard sensitivity analysis could be employed. The approach used is in reality a local sensitivity analysis where one variable is varied while keeping the rest constant. This works for assessing the sensitivity at one location in the parameter space, hence “local”. There are metrics that can be calculated for sensitivity analysis that should be used. The authors should also consider other (global) sensitivity methods to get a handle on the uncertainty. In any case, an established sensitivity metric is a requirement, as opposed to the ad hoc approach reported in Table S10. In addition, the sensitivity analysis does not address the sensitivity in the parameters for TS or for the TS models itself, probably underestimating the variance estimate of the abundance.	We have added a global sensitivity analysis (GSA) to analysis and highlight this in our main methods section as outlined in Lines 187-189 and our results are updated to reflect this. The GSA involved 2000 random permutations amongst the TS of each of the eleven fish taxa. The GSA (from our refitted GAMM) revealed that the mean biomass estimated was lower but still considerably larger than net based estimates. The GSA also demonstrates a high degree of uncertainty Table 2 – which we clearly report in our main results section (Lines 210-218) and discussion Lines 266-268. We highlight that such a high degree of variation and uncertainty is entirely to be expected, as TS of fish is on a logarithmic scale. Given that our (and all acoustic biomass calculations) involve the conversion of data from log to linear scales, we would expect order of magnitude differences in the range of biomass values potentially estimated. However, our biomass estimates based on the interquartile ranges are the best available current estimates. In addition, our paper provides the biological, ecological and acoustic community with empirical measurements of density. We retained the krill contribution to backscatter as a separate sensitivity analysis, as the krill density estimates were assumed to be final krill density. However, this was also performed with random permutation (Lines 189-195), with main results now in Table 2 and additional detail in supplementary information. We have also updated the figure (S5) to clarify the unaccounted backscatter is the addition of krill.
1.7 The authors (in the abstract) claims that the TS of dominant mesopelagic fish in the Scotia Sea is estimated for the first time. The authors have used a simplified model (finite cylinder) for non-swimbladdered fish and have not validated their estimated TS against any field measurements. Although the approach is appropriate for the methods used in the paper, they cannot claim that they have provided a reliable TS model/estimate for the mesopelagic fish. The same concerns exist for swimbladdered mesopelagic fish.	Line 14 has been rephrased from ‘Here, we derive the first acoustic target strength estimates for the dominant mesopelagic fish of the Scotia Sea in the Southern Ocean’ to ‘Here, we use empirical data to estimate species-specific acoustic target strength for the dominant mesopelagic fish of the Scotia Sea in the Southern Ocean.’

1.8 The authors claim that they “...did not include any resonance effect in the TS models...” (Line 129). The models described in equations S11-14 are resonant backscattering models even if the resonance frequency occur lower than 38 kHz. For several smaller size groups, especially at deeper waters, the resonant scatterers dominates the backscatter. Authors need to provide evidence that smaller gas-bearing organisms are properly sampled, or, if not, include that in the sensitivity analysis.

The reviewer is correct in that the equations S10-S14 are resonance models. We also recognise that when interpreting acoustic data, resonance from smaller gas-bearing organisms can dominate backscatter. However, we did not include any resonance effect in our biomass calculation for a number of reasons: (i) as highlighted by the reviewer we used resonance models to explore the potential for smaller gas-bladders to be contributing to the signal and concluded that as the fish in our survey were predominantly beyond resonating size, it was likely to be of limited impact, (ii) the majority of the mesopelagic taxa in the Scotia Sea, where the acoustic data was sampled, are adults as the community is understood to be supported by mass immigration (Saunders et al. 2017), (iii) we are predicting biomass from modelled NASC, this modelled data is likely to have any transient resonance peaks removed as multiple years and seasons are used to model NASC over a broad spatial scale.

We have rephrased **Line 143** to read “...did not include any resonance effect in the biomass calculations as the estimated gas radius of the species in this study was greater than 1 mm (see electronic supplementary material and figure S1), and the myctophid fish community of the Scotia Sea is dominated by expatriates so juvenile fish are less likely to contribute to the signal [29]. In addition, the process of predicting NASC would have smoothed resonance peaks as multiple years of data were used to train the GAMM and predict NASC.”

Saunders RA, Collins MA, Stowasser G, Tarling GA. 2017 Southern Ocean mesopelagic fish communities in the Scotia Sea are sustained by mass immigration. *Mar. Ecol. Prog. Ser.* **569**, 173-185. (doi:10.3354/meps12093).

Reviewer 2	Response
2.1 The authors report on an approach to estimate the cryptic biomass of mesopelagic fish in the Southern Ocean. By using some generic, but Antarctic specific, estimates of acoustic properties and basic shapes of dominant mesopelagic fishes, compositional data from some net tows collected in several areas of the Atlantic Sector of the Southern Ocean, acoustic data (38 kHz) taken along transects over a six year period, and climatological satellite-based data, the authors estimate that krill biomass in the Southern Ocean is between 300 and 700 million tonnes, or up to 70% of the most recent global estimate. The paper is well-written and clear. The authors do a good job of describing the caveats of the work they have produced. However, uncertainty (rather than calculated range) of estimates is really not well described. More thought and specific need to be incorporated within the discussion (see below). This paper needs some additional specificity in the discussion, and as such should be considered for publication after considering the following comments.	As also highlighted by Reviewer 1, we acknowledge that uncertainty was not well described. We have added additional sensitivity analysis to the SI (see responses to Reviewer 1), where we describe the uncertainty in our modelled biomass estimates.
2.2 Discussion Line 270-274 lines to 272-274. While it is true that the TS scaling the biomass is probably the single biggest issue for uncertainty in the definition of an acoustic model for the different species, and the proper weighting of mesopelagic fish biomass in a sampled area, uncertainty regarding the spatial model used to extrapolate the data from the Atlantic to the remainder of the Southern Ocean using the GAMM is probably a bigger over uncertainty to the total biomass. See Figure 2. Although the authors argue that the model is a good fit, it really is not well constrained, and there is no evidence that the surface features (seasonal SST) are really properly describing the subsurface features that define the boundaries of different communities around the Southern Ocean. For example there are areas where seasonal average SST does not match the position of fronts as defined by Orsi (and visualized in this paper). As the seasonal SST and the Orsi fronts describe different dynamical aspects of the ocean, the difference between the two and its implications on estimating biomass are important.	We understand the reviewers concern about an apparent disconnect between peak biomass and sub-surface features. However, the seasonal sea ice extent (also used here to predict NASC) is likely to be vital in the spatial structuring of the fish that were the focus of our study, as while some of the taxa are polar high latitude specialists, none are known to be truly cryopelagic. We have highlighted the influence of the ice edge in Line 227-229 “These biomass peaks are prominent towards the seasonal ice-edge (figure S7), in regions where SST is low (≤ 1 °C), and dominated by non-gas bearing species...” and also raise this in the discussion Lines 319-324. In addition, rather than being single fixed structures, Southern Ocean fronts are known to be highly dynamic and multi streamed, with the position and dynamics of Southern Ocean fronts far from agreed and an active area of research (Chapman et al. 2020). Armour et al. (2016) summarises that SST patterns are mirrored by trends in depth integrated heat content. Subsurface temps are also sparse for the SO, particularly south of the ACC, hence the use of surface products are essential to large scale modelling. Chapman CC, Lea M-A, Meyer A, Sallée J-B, Hindell M. 2020 Defining Southern Ocean fronts and their influence on biological and physical processes in a changing climate. Nat. Clim. Chang. 10(3), 209-219. (doi:10.1038/s41558-020-0705-4). Armour KC, Marshall J, Scott JR, Donohoe A, Newsom ER. 2016 Southern Ocean warming delayed by circumpolar upwelling and equatorward transport. Nat. Geosci. 9, 549–554. (doi:10.1038/ngeo2731).

2.3 If the 700 million tonne value is to be believable, the authors should provide some calculations that the system can actually support an equal biomass of Antarctic krill and of mesopelagic fish.	Following model revision and sensitivity analysis our revised estimates are now lower (though at ~570Mt still appreciably larger than previous net based estimates). We have also provided some simple calculations to the SI to demonstrate that primary production in the Southern Ocean is sufficient to support this level of fish biomass alongside krill. We refer to these in Lines 195-196 “We also used equations in [45], to confirm that annual primary production in the region could support the levels of fish biomass estimated by our model.” and Discussion Lines 279-281 “While our fish biomass estimate is relatively high in comparison to global net based estimates of 1,000 Mt [1], a simple calculation [45] indicates that an annual primary production of 54 g C m⁻²a⁻¹ [52] is sufficient to sustain the requirements of such high levels of fish and krill at 29.4 g C m⁻²a⁻¹.”
2.4 Line 275 to 288 Next lines from 275 to 288 the authors quite nicely note the differences in the selectivity and availability of different species to the nets that have been used. With a larger number of nets tows the probability of missing taxa could be estimated, but in this case it is likely that the compositional changes and dominance are probably the biggest issues, rather than the possibility of missing some species. This the allocation of energy among species with similar (+/- 2dB) TS might be a bigger issue. Of course the total lack of samples from the Pacific is a large uncertainty, as noted. The hypothesis that community and biomass density of an entire ocean basin that is > 50% of the Southern Ocean is similar enough that the SST can provide an adequate constraint is difficult to understand. The Scotia Sea and Southwest Atlantic has the highest primary production of the Southern Ocean, and likely can support more biomass. What is the difference in mean summer primary production in these areas... could the biomass be constrained by the differences in primary production.	Whilst primary productivity was shown to be a good predictor of acoustic backscatter in previous studies, these were focused on more temperate latitudes (Irigoien et al. 2014). Temperature has been shown to be a dominating driver for species distributions in the Southern ocean (Freer et al. 2019), with primary productivity important to a lesser extent and only important for a limited number of taxa (mostly E. carlsbergi). We had included chlorophyll as a proxy for primary productivity in our model, and then excluded it as it had limited effect on adjusted R². However, we were keen to explore this further. We re-ran our model selection, replacing chlorophyll with a climatology of net primary productivity (NPP) to differentiate high and low production areas. While NPP remained of limited influence it did improve model performance, so we retained it along with geostrophic current speed, selecting a final GAMM based on a combination of low AIC, BIC and high R² values. While the new model structure does lower our biomass estimates it does not change the distribution pattern. We have also performed a simple calculation to confirm that Southern Ocean primary productivity can support our estimate of fish biomass alongside krill biomass. See response to point 2.3 and SI section “Primary production required to support biomass” Freer JJ, Tarling GA, Collins MA, Partridge JC, Genner MJ. 2019 Predicting future distributions of lanternfish, a significant ecological resource within the Southern Ocean. Divers. Distrib. 25(8), 1259-1272. (doi:10.1111/ddi.12934). Irigoien X, Klevjer TA, Rostad A, Martinez U, Boyra G, Acuna JL, Bode A, Echevarria F, Gonzalez-Gordillo JJ, Hernandez-Leon S, et al. 2014 Large mesopelagic fishes biomass and trophic efficiency in the open ocean. Nat. Comm. 5, 3271. (doi:10.1038/ncomms4271).

2.5 Lines 297 - 298 be more specific about what the climatology is.. Is this SST?	Rephrased line 317 from - ‘...and climatologies were averaged over multiple years and seasons ...’ to read ‘...and the climatologies used to predict NASC (SST, primary productivity, geostrophic current speed, daylight hours and sea ice concentration) were averaged over multiple years and seasons...’
2.6 Line 300 to 311 this first paragraph is rather specific to South Georgia and to King penguins in that area. Are the spatial patterns likely to really mostly effect king penguins... The location of production around the Antarctic in these fronts could be extremely important for overwinter migrations by chinstrap penguins that have been observed to migrate from the Peninsula to nearly the Ross Sea to the west along the ice edge. It seems like some taxa might be aided by the southern movement of the fronts in the future, although, the authors have not indicated how far these features are likely to move (e.g. 50km average, 100 km, 250km) and this makes difficult to evaluate the conclusions that such changes will result in profound effects (line 317).	Our discussion on ecosystem implications focused on King penguins and seals as myctophids (the dominant mesopelagic fish taxa) constitute the main part of their diets, whereas chinstrap penguin diet is dominated by krill. While we agree that quantifying future changes to the distribution of mesopelagic fish is instrumental to understanding the impact on the top predators, this was not the aim of our current study, which was to develop an acoustic biomass estimate for mesopelagic fish in the Southern Ocean.
2.7 Line 312-322 the authors are rather superficial in the discussion of the impacts to mesopelagic fish resources. What are the likely changes in sub-surface temperatures in the vast majority of the system, how much has it changed in the last 50 years, if projected to the next 50, would the species response be observable?	We agree that changes in past and future environmental conditions are interesting avenues for research, which we plan to address in a future publication. However they are beyond the scope and aims of our current paper. Our comments related specifically to the work of several authors that predicts both poleward shift in species distributions (Freer et al. 2019) and an increase in acoustic backscatter (Proud et al. 2017). With this paragraph we are particularly highlighting that any predicted increase in acoustic backscatter may signify a reduction in fish biomass (rather than an increase as had been presumed). Freer JJ, Tarling GA, Collins MA, Partridge JC, Genner MJ. 2019 Predicting future distributions of lanternfish, a significant ecological resource within the Southern Ocean. Divers. Distrib. 25(8), 1259-1272. (doi:10.1111/ddi.12934). Proud R, Cox MJ, Brierley AS. 2017 Biogeography of the global ocean's mesopelagic zone. Curr. Biol. 27(1), 113-119. (doi:10.1016/j.cub.2016.11.003).
2.8 The argument that the acoustic signal may provide an early indication of structural change is interesting. However what is the current inter-annual or inter-seasonal variability in the acoustic signals in different regions. How long a time series would be required to see a 10% decline in TS that would result in a defined percentage change in biomass. Would this be the same farther south from the principal sampling locations. Would this be the same in the Pacific where primary production is also likely to change with warming and climate change. I would challenge the authors to go further to constrain the effects. Otherwise the results of this paper will either be cited as a major result, or derided as too speculative.	The reviewer raises an interesting point regarding backscatter variability. Currently we lack sufficient regular time-series at the seasonal, annual, or spatial scale to answer this question, hence the use of predictive models. However we have highlighted this in our closing statement while proposing a way forward. Lines 348-351 “While we currently lack sufficient regular acoustic time-series to monitor this change, the increasing use of ships of opportunity, moored echosounders and autonomous vehicles to collect calibrated acoustic data could help build the time-series required and fill knowledge gaps in data poor regions.”

Figure 2

Figure 2. The high biomass of fish predicted by the model is interesting, However in Figure 2 there seems to be a large discrepancy between the location of the fronts that drive the biomass and the seasonal SS. This is especially noticeable in the Indian Ocean sector where the high biomass is associated with SST gradient, but is well north of the position of the Southern Antarctic Circumpolar Current Front. This disconnect also occurs in the Southeast Pacific near the West Antarctic peninsula. So it would be interesting to understand these difference impact the likely outcomes of the model. To this reviewer, these differences suggest a disconnect between the oceanography that generates the subsurface features defining the spatial habitat, and the surface expression of a features that are weakly correlated with the dynamics in some areas. The authors should consider what impact this could have on their estimates.

See response to 2.1

Appendix B

Response to comments 23-Dec-2021

Comment	Response
The authors have largely addressed the reviewers comments in revising their manuscript. However, there are a couple of points raised by the reviewers that the authors have seemingly passed over superficially and could be more fully addressed with the addition/revision of some of the content in the discussion in identifying the uncertainties of the study, which both authors are calling to be more adequately addressed. I would consider that this would only entail a sentence or two and these minor considerations will strengthen the manuscript and better respond to the reviewers comments.	We thank the editor for these useful suggestions and have responded as follows.
In particular, the second reviewer raised the issue of the use of surface features in describing complex subsurface features and how this might contribute to uncertainty in modelled outputs. The authors have passed over this comment by identifying that the reviewers reference to Orsi was not necessarily appropriate and referencing one publication to provide evidence that surface features adequately reflect subsurface features. Given the complexity of features associated with frontal regions, varying views on understanding of these features and how they are changing both directly as a result of increased carbon uptake and also changes in climate phenomena that influence them (see the IPCC's SROCC), I'd suggest that the authors include some small revisions to the discussion identifying some of the limitations associated with use of SST to describe complex subsurface features (particularly when studying biological features that only spend part of their day in the upper parts of the ocean). They may want to discuss this in light of the correlations between SST and SST200 and why SST might be considered more appropriate. This will strengthen the manuscripts consideration of uncertainty. Further, the second reviewer queried the disconnect between the areas of high biomass estimated by the model and the location of fronts, with the areas of high biomass north of frontal averages. The authors have responded with a justification centered around the extent of sea ice. While I agree that sea ice extent will no doubt have some influence on distributions of species that are not associated with the cryosphere, I'm not sure I fully understand this justification given the northward location of the high biomass (and sampling outside of the maximum sea ice extent), so in regions beyond the relevant seasonal northward limits of sea ice extent. In considering oceanographic features that might influence species compositions and distributions more broadly, authors might want to consider the higher rate of ocean warming northward of the ACC and the oceanographic processes contributing to this (again see the IPCC SROCC) and what influences these changes might be having on productivity in these northern regions of the Southern Ocean rather than solely focusing on sea ice extent. As identified above this would only require minor revision to the discussion and would strengthen the discussion on uncertainty.	We have included revisions in the main manuscript, Lines 322-342 (Main Document) to incorporate both the limitations and benefits of using SST, and identify warming in the northern ACC and implications for biomass estimates and uncertainty in address these comments.
In addition I have attached some minor editorial revisions for consideration by the authors in finalising their manuscript.	We have accepted and/or modified suggestions throughout. We have clarified Lines 18-20 and 253-254 to explicitly indicate the source of biomass estimates.